# Introducing Hann windows for reducing edge-effects in patch-based image segmentation

**Nicolas Pielawski**[ID][1]*, **Carolina Wählby**[ID][1,2]*

**1** Department of IT, Uppsala University, Uppsala, Sweden, **2** BioImage Informatics Facility of SciLifeLab, Uppsala, Sweden

* nicolas.pielawski@it.uu.se (NP); carolina.wahlby@it.uu.se (CW)

**Data Availability Statement:** https://github.com/easycui/nuclei_segmentation/tree/master/datasets/Multi_organs.

**Funding:** This project was financially supported by the Swedish Foundation for Strategic Research

## Abstract

There is a limitation in the size of an image that can be processed using computationally demanding methods such as e.g. Convolutional Neural Networks (CNNs). Furthermore, many networks are designed to work with a pre-determined fixed image size. Some imaging modalities—notably biological and medical—can result in images up to a few gigapixels in size, meaning that they have to be divided into smaller parts, or patches, for processing. However, when performing pixel classification, this may lead to undesirable artefacts, such as edge effects in the final re-combined image. We introduce windowing methods from signal processing to effectively reduce such edge effects. With the assumption that the central part of an image patch often holds richer contextual information than its sides and corners, we reconstruct the prediction by overlapping patches that are being weighted depending on 2-dimensional windows. We compare the results of simple averaging and four different windows: Hann, Bartlett-Hann, Triangular and a recently proposed window by Cui et al., and show that the cosine-based Hann window achieves the best improvement as measured by the Structural Similarity Index (SSIM). We also apply the Dice score to show that classification errors close to patch edges are reduced. The proposed windowing method can be used together with any CNN model for segmentation without any modification and significantly improves network predictions.

## Introduction

Semantic image segmentation is a process consisting of separating an image into regions, e.g. representing different object types. This problem is ill-defined because there is no general definition of a region, and learning-based methods such as CNNs have started to outperform classical rule-based methods in recent years. In 2015 Ronneberger et al. [1] introduced the U-Net neural network and architecture, consisting of a compressing and decompressing part with skip-connections in between. In 2017, Jégou et al. [2] greatly increased the number of skip-connections used, and reached the state-of-the-art of semantic segmentation with very few trainable parameters, but consequently at the cost of a larger memory footprint.

Segmentation tasks are memory intensive, mainly due to the size of the input images and the preservation of feature maps along the computational graph of a CNN. In some cases, such

(grant SB16-0046 and BD150008) and the European Research Council (ERC-2015-CoG 683810). The funders had no role in study design, data collection and analysis, decision to publish, or preparation of the manuscript.

**Competing interests:** The authors have declared that no competing interests exist.

as when handling giga-pixel-sized whole slide images (WSI) in digital pathology, it is not possible to process the whole image at once [3]. To counteract these memory issues, patch-based segmentation methods use different techniques that feed more or less contextual information to the neural networks. Edge-effects are all known to appear when working with CNNs, and have been approached in different ways: for instance by keeping a contextual border in the input that is removed from the output in the original U-Net architecture. In 2018, Innamorati et al. [4] showed that the errors of segmentation are higher for the pixels near the edges and even worse for the corners. The same year, Cui et al. [5] proposed a method to reduce edge effects and increase the final segmentation quality after reassembling the different patches. Their method consists of weighing the loss function and the patches with a specific mask which will be referred to as Pyramidal window in this paper.

In signal processing, one of the renowned windows is the Hann window, invented by Julius von Hann around 1900 and named after him by Blackman and Tukey [6] in 1958. Window functions are often used to taper a signal, by multiplying a window by a patch extracted from the signal, reducing the importance of the borders. Their usage is broad, for instance when transforming a signal into a spectrogram that can later reconstruct the original signal without artefacts. Another example lies in the field of statistics where curve fitting can be given a weighing factor—the window function—which is also known under the name of kernel. It is desirable for these windows to integrate to one (with a stride of half a window); not filling the criterion requires an additional normalisation step after reconstruction which is the case of the Pyramidal window. This supplementary step can introduce additional artefacts due to rounding errors in floating point arithmetic.

In this paper we explore the idea of windowing to reduce edge effects after CNN-based image segmentation. Our contributions are the following:

- We present a method that can be applied *post hoc* on any CNN segmentation output without needing to retrain or modify the loss function nor renormalise the output.

- We handle edge and corner cases separately, which gives a weighted estimate given the available context. We also avoid additional floating point errors by using windows that integrate to one.

## Methods

The proposed window patch-based method is a refinement step that reduces the edge artefacts at patch borders. Inspired by signal processing, we multiply each patch with a 2-dimensional window function, which gives more emphasis to their centres and less to their adjacent edges and corners. Next, we hypothesise that the most correct information will be kept when combining the window-weighted CNN outputs increasing the quality of our predictions.

Our method follows this pipeline:

1. Extracting overlapping patches, with a stride of half a patch size.

2. Performing the prediction on the patches.

3. Multiplying each patch by the appropriate window depending on its absolute location: the window must be of the same size as the patch and must be replaced if it is associated to a border or corner patch.

4. Summing all the patches at their absolute location.

## Window functions

We evaluate three different windows from classical signal processing: Hann [7], Bartlett-Hann [7], and Triangular [8] and compare with the Pyramidal window [5] as well as with simple averaging of patch overlaps. We chose to focus on evaluating these windows rendered in 2 dimensions. The choice of a window function is arbitrary for as long as it is separable and sums up to 1 when properly integrated over the patches with a stride of half a window size. The Pyramidal window does not follow this requirement and thus needs an extra normalisation step which can introduce additional artefacts.

In signal processing, for an arbitrary 1-dimensional window function $w$, Speake et al. [9] described the 2-dimensional version $W$ in separable form as:

$$W(i,j) = w(i)w(j)$$

We can thus derive the 2-dimensional versions of the original windows as follows:

$$W_{Average}(i,j) = \frac{1}{4} \tag{1}$$

$$W_{Hann}(i,j) = \frac{1}{4} \left( 1 - \cos\left(\frac{2\pi i}{I-1}\right) \right) \left( 1 - \cos\left(\frac{2\pi j}{J-1}\right) \right) \tag{2}$$

$$W_{Bartlett-Hann}(i,j) = \quad \left( a_0 + a_1 \left| \frac{i}{I} - \frac{1}{2} \right| - a_2 \cos\left(\frac{2\pi i}{I}\right) \right)$$
$$\left( a_0 + a_1 \left| \frac{j}{J} - \frac{1}{2} \right| - a_2 \cos\left(\frac{2\pi j}{J}\right) \right) \tag{3}$$

$$W_{Triangular}(i,j) = \left( 1 - \left| \frac{2i}{I} - 1 \right| \right) \left( 1 - \left| \frac{2j}{J} - 1 \right| \right) \tag{4}$$

where $i$ the current horizontal position, $I$ the width of the patch, $j$ the current vertical position, $J$ the height of the patch, $a_0 = 0.62$, $a_1 = 0.48$, and $a_2 = 0.38$ [7].

Cui et al. [5] defined a weighted loss function which is later used as a window. The resulting Pyramidal window is defined as follows:

$$W_{Pyramidal}(i,j) = \alpha \frac{D^e_{i,j}}{D^c_{i,j} + D^e_{i,j}} \tag{5}$$

$$\alpha = \frac{I \cdot J}{\sum_{i=1}^{I} \sum_{j=1}^{J} \frac{D^e_{i,j}}{D^c_{i,j} + D^e_{i,j}}}$$

where $D^e_{i,j}$ the absolute distance from the edge and $D^c_{i,j}$ the distance from the centre. The Pyramidal window formula above cannot be described in separable form which consequently increases the cost of computing the 2-dimensional window.

The 2-dimensional realisation of the different windows can be seen in Fig 1.

**Complexity.** The complexity of a non-overlapping reconstruction takes $nm$ computations of individual patches, with $n$ the number of patches horizontally, and $m$ the number of patches

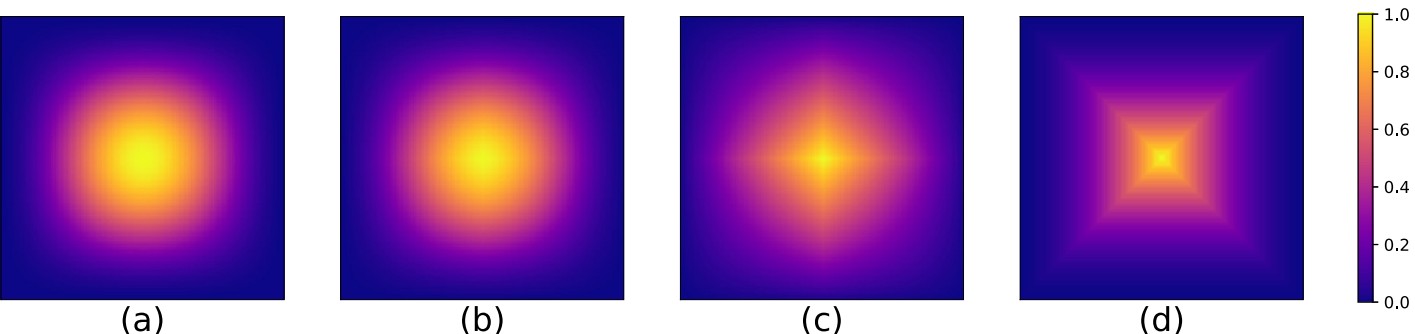

**Fig 1. Illustration of the different 2-dimensional windows.** Every window gives more emphasis to the information in the centre than the information on the borders and in corners. **(a)** 2D Hann window, **(b)** 2D Bartlett-Hann window, **(c)** 2D Triangular window, and **(d)** 2D Pyramidal window (normalised between 0 and 1).

vertically. Our method has a complexity of

$$(2n - 1)(2m - 1) = 4nm - 2n - 2m + 1$$

That is approximately $4nm$, and this approximation becomes more accurate as $n$ or $m$ grows, i.e., the image gets larger.

In practice, the non-overlapping reconstruction can be performed first in order to display a preview to a user, and then complete the missing information by computing the three quarters of the remaining patches in the meanwhile. Another optimisation is to combine both methods and use the fast non-overlapping reconstruction method for the inessential details of an image (e.g. the background) and the windowed method for the objects of interest.

## Edges and corners

Contextual information from adjacent patches is naturally not available at the edges and corners of the full-size image. Therefore we propose specific windows to increase inference accuracy in these parts of the final image. The patches nearby the edges and corners are weighted with a different set of windows to compensate for the missing information so that the sum of all overlapping windows over the full image is 1. The 2-dimensional border windows of an image can be constructed from an arbitrary 1-dimensional window $w$:

$$W_{Up}(i,j) = \begin{cases} w(i), & \text{if } j < \frac{l}{2} \\ w(i)w(j), & \text{otherwise} \end{cases} \tag{6}$$

$$W_{Down}(i,j) = \begin{cases} w(i), & \text{if } j > \frac{l}{2} \\ w(i)w(j), & \text{otherwise} \end{cases} \tag{7}$$

$$W_{Left}(i,j) = \begin{cases} w(j), & \text{if } i < \frac{l}{2} \\ w(i)w(j), & \text{otherwise} \end{cases} \tag{8}$$

$$W_{Right}(i,j) = \begin{cases} w(j), & \text{if } i > \frac{l}{2} \\ w(i)w(j), & \text{otherwise} \end{cases} \tag{9}$$

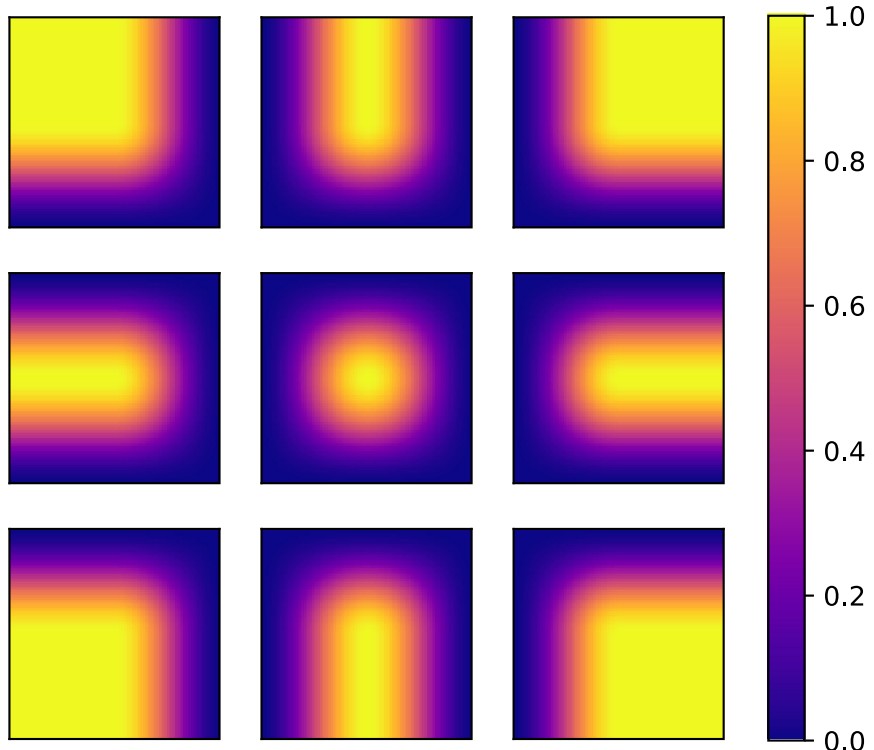

**Fig 2. The different configurations of Hann windows of size 128x128 for edge and corner cases.** Summing all the windows with the appropriate amount of overlap ($\frac{128}{2} = 64$ pixels in this example) makes every window applied to the reconstructed image sum to 1.

with $W(i, j)$ the resulting border window, $w(i)$ the evaluation of an arbitrary 1-dimensional window $w$ at position $i$.

The formula of the upper left corner patch of an image is defined as:

$$
W_{UpLeft}(i,j) = \begin{cases} 1, & \text{if } i \leq \frac{l}{2} \text{ and } j \leq \frac{l}{2} \\ w(i), & \text{if } i > \frac{l}{2} \text{ and } j < \frac{l}{2} \\ w(j), & \text{if } i < \frac{l}{2} \text{ and } j > \frac{l}{2} \\ w(i)w(j), & \text{otherwise} \end{cases} \tag{10}
$$

It is possible to construct the three remaining corner windows from 10. These formulas yield eight new windows as visualised in Fig 2.

## Experiments

We present an experiment where six different methods for combining the results of patch-based image segmentation are compared. The experiments follow the same protocol and data as in Cui et al. [5], where a U-Net neural network architecture is trained on a hematoxylin and eosin (H&E)-stained tissue dataset. The model is trained with a cross-entropy loss, and the weighting loss as constructed by Cui et al. [5] was discarded. The patch size is 128x128 for a complete image of 1024x1024 pixels. The six different methods that are compared are: No overlap, Average, Pyramidal, Hann, Bartlett-Hann, and Triangular windows.

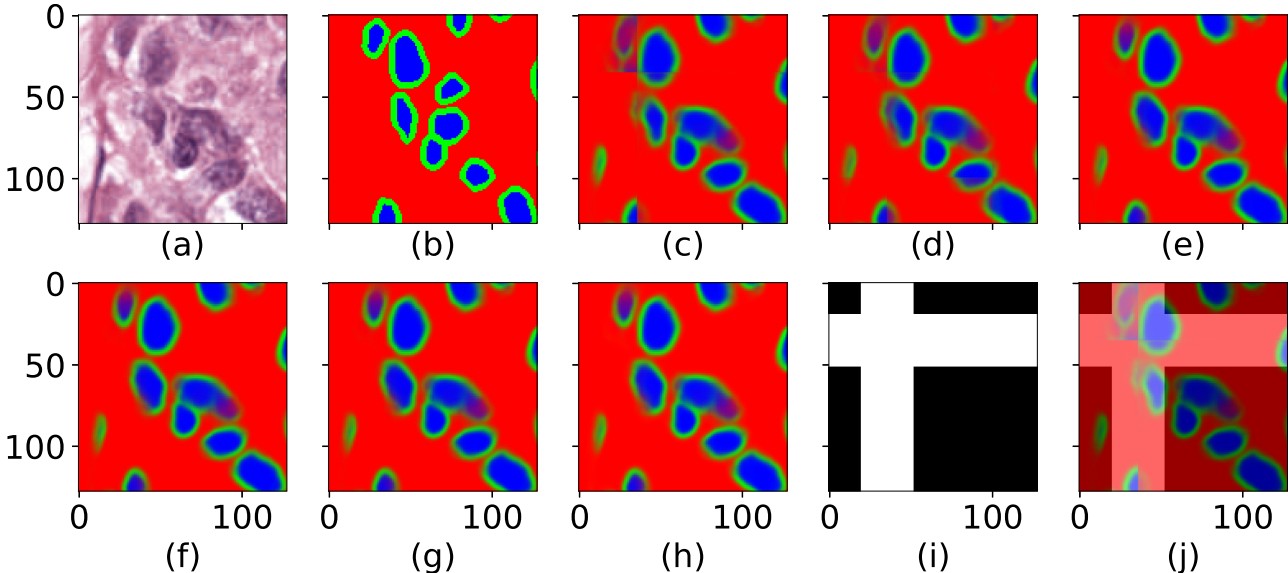

**Fig 3. Input, ground-truth and different reconstruction methods.** The figure is cropped from 1024x1024 to 128x128 for visibility. **(a)** The input colour image. **(b)** The ground-truth. **(c)** The baseline, where the patches are assembled without overlaps, "No overlap". **(d)** The overlapping patches assembled with an average window. **(e)** The patches assembled and weighted with a Pyramidal window. **(f)** The patches assembled and weighted with a Hann window. **(g)** The patches assembled and weighted with a Bartley-Hann window. **(h)** The patches assembled and weighted with a Triangular window. **(i)** The binary mask used to define patch edges and vicinity of edges as used for calculation of Dice coefficients, and **(j)**, an overlay of the mask and (c).

Fig 3(a)–3(h) show input, ground-truth, and examples of results of the different methods. Although subtle, artefacts are most visible in the "No overlap" and the Average windowing, while the visual artefacts after reconstruction with the proposed windowing approaches are marginal.

Here, we focus on evaluating the benefit of using windowing when combining patches of a U-Net output. However, the ground truth we have access to consists of manual annotations describing the desired U-Net output. This means that any direct comparison to ground truth will be a mixture of U-Net performance and windowing effects. We therefore chose two strategies for evaluation. First we compare ground truth (exemplified in Fig 3(b)) and U-Net output without reconstruction (No overlap, exemplified in Fig 3c) as well as the different reconstructed outputs (exemplified in Fig 3(d)–3(h)) using using the structural similarity index [10] (SSIM). Next we compare to the ground truth using the Dice coefficient, and to separate U-Net performance errors from windowing effects, we measure the Dice coefficient separately for pixels in patch centres and pixels in the vicinity of edges, using a binary mask as described below.

The SSIM is a method used to measure the similarity between two images, and is related to the human visual perception. Moreover, the SSIM generates statistics from local structures by using a window in order to compute a score, in opposite to single pixel methods such as PSNR (Peak Signal-to-Noise Ratio). The U-Net prediction consists of three classes, and we calculated the SSIM for each of the three classes, and then average in order to achieve the resulting scores. As pointed out above, the produced windowing reduces artefacts which are present on only a small number of pixels at the edges of patches. This results in a low variance of the SSIM scores. To better visualise the effects of windowing we subtract the SSIM of the baseline with the most prominent artefacts, i.e. the result of not having patch overlap (No overlap), from the SSIM of each method. Fig 4(a) and 4(b) shows the adjusted SSIM for each windowing

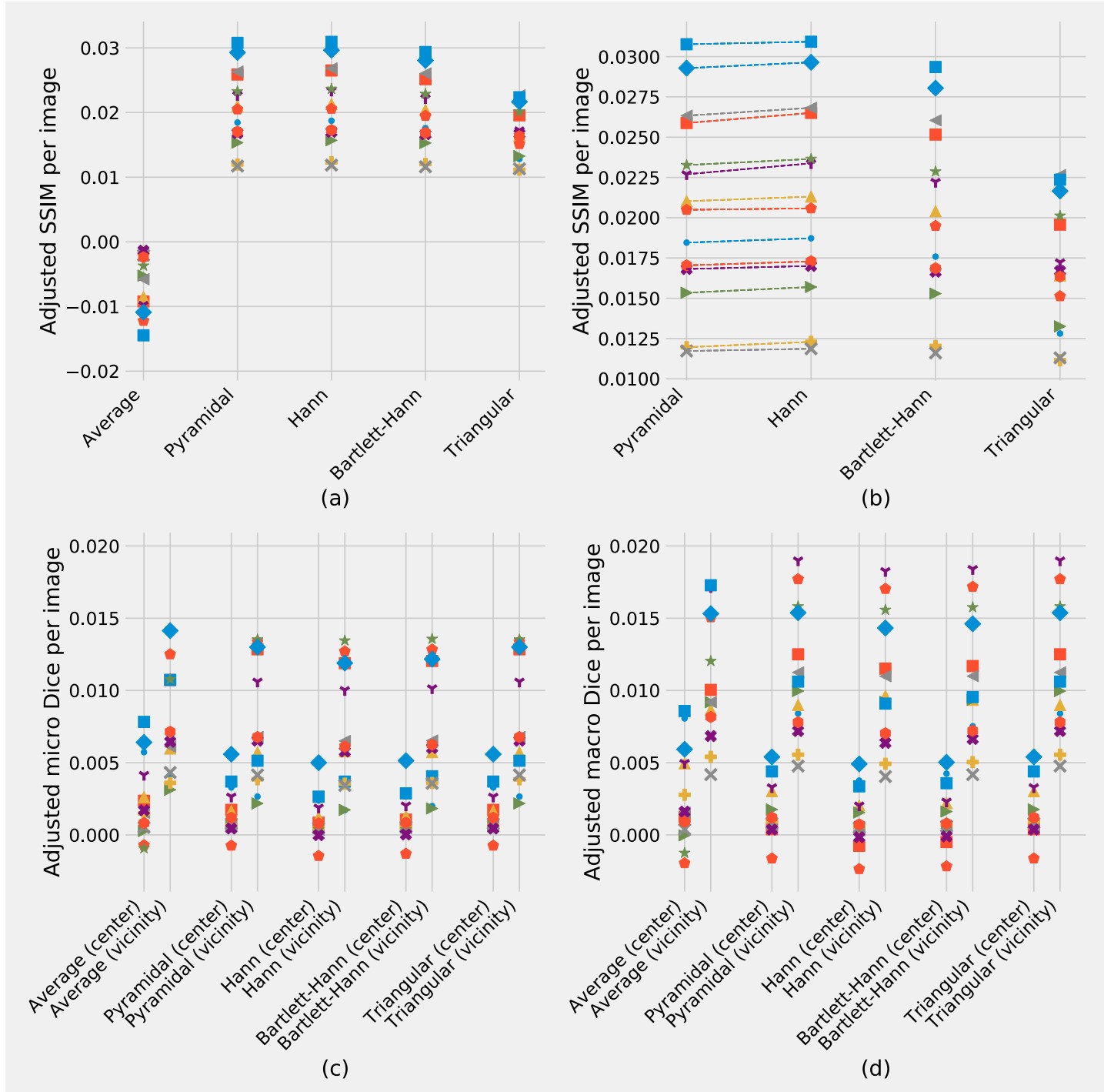

**Fig 4. Comparison of windowing approaches using SSIM and Dice coefficients. (a)** and **(b)** show SSIM for five different windowing methods as adjusted to a baseline, which is the SSIM from "No overlap" patches. The SSIM indexes have been computed class-wise and then averaged. **(b)** shows a zoomed in version of (a) excluding the average window in order to emphasise the differences. The differences between the Pyramidal and the Hann windows are not clearly visible and dashed lines have been plotted to display a visual trend. **(c)** and **(d)** show the micro and macro Dice coefficients respectively, after adjusting (by subtraction) for the corresponding Dice coefficients of "No overlap" patches, to separate windowing errors from U-Net errors. The Dice coefficients were calculated separately for patch centres and edge vicinity, showing how the windowing has less of an effect in patch centres but improves the Dice coefficient at edge vicinity. Each of the 14 sample images has a unique combination of symbol and colour.

approach and each image. Each marker is representing an image with a unique combination of symbol and colour.

The Dice coefficient is typically used to compare segmentation masks, and as we have three classes we calculated both micro and macro Dice [11]. The micro-averaging and macro-averaging of the Dice coefficients were computed following the definition of Sebastiani [12, p. 33]. We started from the reconstructed U-Net outputs and assigned each pixel its most probable class, resulting in three binary images representing objects (blue), object edges (green), and image background (red). Next, the micro and macro Dice coefficients comparison to the ground truth was calculated separately for pixels in patch centres and pixels in the vicinity of edges using a binary mask as shown in Fig 3(i) and overlaid in Fig 3(j). As with the SSIM comparison, we adjust the scores by subtracting the result of not having patch overlap (No overlap), from the Dice score of each method. Fig 4(c) and 4(d) show the adjusted micro and macro Dice score for patch centres and edges vicinity for each windowing approach and image.

## Hypothesis testing

Besides the average window, all windows' SSIM significantly outperformed the baseline (using adjacent patches, without overlaps, i.e. "No overlap"). An independent paired t-test was conducted and showed evidence that the Pyramidal window (mean: $0.02079 \pm 0.00597$; $t(13) = 13.0294$, $p < .0001$), Hann window (mean: $0.02112 \pm 0.00603$; $t(13) = 13.1095$, $p < .0001$), Bartlett-Hann window (mean: $0.02026 \pm 0.00562$; $t(13) = 13.4807$, $p < .0001$) and Triangular window (mean: $0.01691 \pm 0.00393$; $t(13) = 16.0797$, $p < .0001$) all yielded a better score than the baseline.

Surprisingly, predictions with overlapping patches weighted by an average window (mean: $-0.00710 \pm 0.00447$) performed worse than our baseline; $t(13) = 5.9431$, $p < 0.0001$, even though they contain about four times as much information. This most likely happens because the overlapping patches do not weigh down edge artefacts thus yielding four times as many artefacts.

In our results, the Hann outperformed the Pyramidal window by a small amount and we performed an exact sign test [13] in order to highlight those differences in SSIM. The Hann window elicited a statistically significant mean increase in SSIM ($0.00033 \pm 0.00017$) compared to the Pyramidal window; $t(13) = 7.0849$, $p < .0001$.

In terms of Dice coefficient, all methods significantly outperformed the baseline, including the prediction with overlapping patches weighted by an average window (see Fig 5(a)). The triangular and pyramidal window got exactly the same Dice coefficients for all images, and significantly outperformed other methods, except for the overlapping patches. As stochasticity is added when computing the argmax of the softmax, it becomes more difficult to compare the Dice coefficient of the different methods (see Fig 5(b)).

## Discussion and future work

Our method offers an immediate reduction in edge artefacts as assessed with SSIM and can be easily implemented and integrated without the need of modifying any existing Deep Learning model. Moreover, signal theory provides a basis for further improvements. Our results suggest that using a Hann window is an effective way of reducing edge artefacts, and testing the method on different datasets and image modalities could further confirm our hypothesis. More window types could be compared and a more in-depth study could focus on determining the best window type depending on the circumstances.

| Paired t-test | | Method A | Method B | μ | s/√n | t-value | | p-value |
|---|---|---|---|---|---|---|---|---|
| Micro-Dice | Center | Overlaps | Baseline | 0.00242 | 0.00267 | t(13)=3.3793 | ● | 0.0050 |
| | | Pyramidal | Baseline | 0.00161 | 0.00167 | t(13)=3.6237 | ● | 0.0031 |
| | | Hann | Baseline | 0.00111 | 0.00152 | t(13)=2.7318 | ● | 0.0172 |
| | | Bartlett-Hann | Baseline | 0.00122 | 0.00156 | t(13)=2.9437 | ● | 0.0115 |
| | | Triangular | Baseline | 0.00161 | 0.00167 | t(13)=3.6237 | ● | 0.0031 |
| | Grid | Overlaps | Baseline | 0.00789 | 0.00362 | t(13)=8.1537 | ● | 0.0001 |
| | | Pyramidal | Baseline | 0.00765 | 0.00417 | t(13)=6.8658 | ● | 0.0001 |
| | | Hann | Baseline | 0.00702 | 0.00417 | t(13)=6.2931 | ● | 0.0001 |
| | | Bartlett-Hann | Baseline | 0.00717 | 0.00417 | t(13)=6.4356 | ● | 0.0001 |
| | | Triangular | Baseline | 0.00765 | 0.00417 | t(13)=6.8658 | ● | 0.0001 |
| Macro-Dice | Center | Overlaps | Baseline | 0.00270 | 0.00330 | t(13)=3.0541 | ● | 0.0093 |
| | | Pyramidal | Baseline | 0.00188 | 0.00207 | t(13)=3.3913 | ● | 0.0049 |
| | | Hann | Baseline | 0.00121 | 0.00191 | t(13)=2.3785 | ● | 0.0334 |
| | | Bartlett-Hann | Baseline | 0.00137 | 0.00195 | t(13)=2.6380 | ● | 0.0205 |
| | | Triangular | Baseline | 0.00188 | 0.00207 | t(13)=3.3913 | ● | 0.0049 |
| | Grid | Overlaps | Baseline | 0.01049 | 0.00423 | t(13)=9.2772 | ● | 0.0001 |
| | | Pyramidal | Baseline | 0.01107 | 0.00446 | t(13)=9.2811 | ● | 0.0001 |
| | | Hann | Baseline | 0.01036 | 0.00449 | t(13)=8.6299 | ● | 0.0001 |
| | | Bartlett-Hann | Baseline | 0.01053 | 0.00448 | t(13)=8.7919 | ● | 0.0001 |
| | | Triangular | Baseline | 0.01107 | 0.00446 | t(13)=9.2811 | ● | 0.0001 |

(a)

| Paired t-test | | Method A | Method B | μ | s/√n | t-value | | p-value |
|---|---|---|---|---|---|---|---|---|
| Micro-Dice | Center | Overlaps | Triangular | 0.00080 | 0.00134 | t(13)=2.2333 | ● | 0.0438 |
| | | Pyramidal | Triangular | 0.00000 | 0.00000 | - | | - |
| | | Hann | Triangular | -0.00050 | 0.00032 | t(13)=-5.9474 | ● | 0.0001 |
| | | Bartlett-Hann | Triangular | -0.00039 | 0.00024 | t(13)=-5.9770 | ● | 0.0001 |
| | Grid | Overlaps | Triangular | 0.00024 | 0.00193 | t(13)=0.4741 | ◐ | 0.6434 |
| | | Pyramidal | Triangular | 0.00000 | 0.00000 | - | | - |
| | | Hann | Triangular | -0.00063 | 0.00041 | t(13)=-5.7255 | ● | 0.0001 |
| | | Bartlett-Hann | Triangular | -0.00048 | 0.00031 | t(13)=-5.7339 | ● | 0.0001 |
| Macro-Dice | Center | Overlaps | Triangular | 0.00082 | 0.00167 | t(13)=1.8341 | ◐ | 0.0897 |
| | | Pyramidal | Triangular | 0.00000 | 0.00000 | - | | - |
| | | Hann | Triangular | -0.00066 | 0.00043 | t(13)=-5.7560 | ● | 0.0001 |
| | | Bartlett-Hann | Triangular | -0.00050 | 0.00033 | t(13)=-5.6788 | ● | 0.0001 |
| | Grid | Overlaps | Triangular | -0.00057 | 0.00242 | t(13)=-0.8836 | ◐ | 0.3930 |
| | | Pyramidal | Triangular | 0.00000 | 0.00000 | - | | - |
| | | Hann | Triangular | -0.00071 | 0.00051 | t(13)=-5.1273 | ● | 0.0002 |
| | | Bartlett-Hann | Triangular | -0.00053 | 0.00036 | t(13)=-5.5992 | ● | 0.0001 |

(b)

**Fig 5. Paired t-test between the Dice coefficient within grid and outside of grid (patch centres) between the different methods. (a)** Paired t-test between the methods and the baseline (all are statistically significant), **(b)** Paired t-test between the methods and the triangular window.

Assessment of segmentation accuracy, assessed with the Dice coefficient, showed that it is difficult to separate variations in accuracy due to a non-perfect ground truth, a sub-optimal network, and edge effects. We therefore focused on improvements in accuracy at patch edges as compared to patch centres when applying different types of weighted windows, and showed that all versions of weighted windows improved the result. However, the choice of window should be left as a hyper-parameter dependent on the application. Even though the weighting can erase a great deal of information, the predictions at the patch's vicinity was shown to be noisy in [4] and this hypothesis was confirmed by the result of our experiments. Using a window that weighs the centre more than the vicinity of the patch was beneficial to both the SSIM and Dice metrics.

The proposed method assumes that a constant amount of context is needed to have an accurate prediction. It could be of interest to focus on reducing the amount of context needed, which would result in saturating windows. For instance, the Tukey window [8], has a tunable parameter varying the amount of context. This would result in reducing the amount of overlap between the patches and could bring important computational gains.

An optimisation of the context could also be achieved with Deep Bayesian neural networks that yield a prediction associated to an uncertainty. This uncertainty, or precision, could be combined to a Bayesian prior in order to compute a different window for each patch. This window would then be predicted depending on how much context the neural network needs: the more certain a predicted area will be, the more weight it will receive.

## Conclusion

In this paper we describe a new method that introduces Hann windows for reducing the edge-effects when performing image segmentation with CNNs. We explained how to construct arbitrary windows in 2-dimensions and how they can be expanded for borders and corner cases. To demonstrate our concept, we tested six different windows on a cell nuclei dataset and showed that it compares favourably with an existing method provided by Cui et al. [5]. Finally, the method is readily available and simple to implement in existing Deep Learning models, even if they are already trained.

## Author Contributions

**Conceptualization:** Nicolas Pielawski.

**Formal analysis:** Nicolas Pielawski.

**Funding acquisition:** Carolina Wählby.

**Investigation:** Nicolas Pielawski, Carolina Wählby.

**Methodology:** Nicolas Pielawski, Carolina Wählby.

**Project administration:** Carolina Wählby.

**Resources:** Carolina Wählby.

**Software:** Nicolas Pielawski.

**Supervision:** Carolina Wählby.

**Validation:** Carolina Wählby.

**Visualization:** Nicolas Pielawski.

**Writing – original draft:** Nicolas Pielawski, Carolina Wählby.

**Writing – review & editing:** Nicolas Pielawski, Carolina Wählby.

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
