## [Decision Letter · Decision Letter 0]

27 Nov 2019

PONE-D-19-29003

Introducing Hann windows for reducing edge-effects in patch-based image segmentation

PLOS ONE

Dear Mr. Pielawski,

Thank you for submitting your manuscript to PLOS ONE. After careful consideration, we feel that it has merit but does not fully meet PLOS ONE’s publication criteria as it currently stands. Therefore, we invite you to submit a revised version of the manuscript that addresses the points raised during the review process.

Please revise the manuscript by considering the reviewers' comments.

We would appreciate receiving your revised manuscript by Jan 11 2020 11:59PM. To enhance the reproducibility of your results, we recommend that if applicable you deposit your laboratory protocols in protocols.io, where a protocol can be assigned its own identifier (DOI) such that it can be cited independently in the future. For instructions see: http://journals.plos.org/plosone/s/submission-guidelines#loc-laboratory-protocols

We look forward to receiving your revised manuscript.

Kind regards,

Jie Zhang

Academic Editor

PLOS ONE

Journal Requirements:

Reviewers' comments:

Reviewer's Responses to Questions

**Comments to the Author**

1. Is the manuscript technically sound, and do the data support the conclusions?

Reviewer #1: Yes

Reviewer #2: Partly

Reviewer #3: Yes

2. Has the statistical analysis been performed appropriately and rigorously? 

Reviewer #1: Yes

Reviewer #2: Yes

Reviewer #3: Yes

3. Have the authors made all data underlying the findings in their manuscript fully available?

Reviewer #1: Yes

Reviewer #2: Yes

Reviewer #3: Yes

4. Is the manuscript presented in an intelligible fashion and written in standard English?

Reviewer #1: Yes

Reviewer #2: Yes

Reviewer #3: Yes

5. Review Comments to the Author

Reviewer #1: The authors present a procedure to address edge effects in CNN image segmentation. Their experiments are documented in detail and the manuscript is in general well-written. I do however have some recommendations that the authors should consider.

References: I find it hard to believe that only 11 references were relevant for this manuscript. The authors should significantly improve their literature review, in order to place their work more rigorously within existing literature.

Computational complexity: Although a mention of the general complexity of their method is shown, it would be very much helpful for the reader to display the computational time for the authors technique – along with the benchmark methods. One of the objectives in the era of big data processing is to find the right balance between improvements in accuracy, and time needed to retrieve the desired outputs and as such, this type of information can be particularly useful.

Validation: I understand why the authors decided to use the SSIM as validation metric. One thing, the SSIM should be presented in detail and not just put in a reference and second, wouldn’t other established accuracy metrics (such as overall accuracy, precision, recall and F1 score) be also fruitful?

Reviewer #2: The paper introduced Hann windows for patch-based image segmentation. The result shows that in terms of Structural similarity index, the Hann windows slightly outperfoms baseline. The novalty of the paper is limited. Eq 6 to Eq 10 are not clear enough, especially the definition of w(i).

Only SSIM is used for evaluation, it is necessary to add commonly used metrics for segmentation tasks, such as DICE.

Minor: Eq 10, should be if i <= L/2 and j <= J/2.

Reviewer #3: This manuscript describes the use of windowing functions on microscope image data that is to be segmented using convolutional neural network (CNN) approaches. In specific, the manuscript compares the effectiveness of several different window functions, including Hann windows. A strength of the approach described in this manuscript is that it provides an effective way to perform CNN-based segmentation on large images by subdividing them into many smaller images, while minimizing any artifacts caused by the segmentation algorithm near the edges of each subdivided image (i.e., edge artifacts). The manuscript is well thought out and well written and only very minimal comments were identified, as described below:

1. With reference to Figure 1 and the color-bar and color look-up tables used in the figure: was a different color look-up table used for the pyramidal window? It was mentioned in the manuscript that the integrated area of the pyramidal window is not 1, while the other window functions do have an integrated area of 1. Was it necessary to compensate for this different integrated area by using a different color look-up table, or alternatively, were the data normalized prior to visualizing as a 2D heatmap?

2. The description of the different approaches and different methods is a bit confusing on page 5, for the top 3 paragraphs. For example, the first paragraph refers to "The five different approaches", while the 3rd paragraph (which is the Fig. 3 caption) refers to "the adjusted SSIM for six different methods". It is a bit unclear of what an "approach" is vs. a "method" and whether there are 5 or 6 of them. In Fig. 3a, there appear to be 5 different window functions shown. For similar reasons, the 2nd paragraph on page 5 is also confusing, in the sentence that reads "Due to a low variance in the SSIM indices for each method but a high variance between methods..." It is unclear here as well what the method is that the sentence is referring to. Clearing up the language in these 3 paragraphs is needed. Finally, along similar lines, the authors might help to clarify the vertical axis of Figure 3 by explaining that this is the improvement in SSIM over baseline, or some wording similar to this.

3. On page 6, first paragraph of the Discussion section, the 2nd sentence contains the word "theory" twice, which is a bit redundant.

6. PLOS authors have the option to publish the peer review history of their article (what does this mean?). If published, this will include your full peer review and any attached files.

Reviewer #1: No

Reviewer #2: No

Reviewer #3: No

---

## [Author Response · Author response to Decision Letter 0]

13 Jan 2020

We thank the reviewers for their constructive comments and have revised the manuscript to address their concerns.

REVIEWER #1

> References: I find it hard to believe that only 11 references were relevant for this manuscript. The authors should significantly improve their literature review, in order to place their work more rigorously within existing literature.

The issue is relatively little discussed as many researchers have access to powerful computers or can reformulate the problem to deal with smaller images. One approach consists of cropping the borders (e.g. Rudolph, Robert, et al. "Efficient identification, localization and quantification of grapevine inflorescences in unprepared field images using fully convolutional networks." arXiv preprint arXiv:1807.03770 (2018)) but can still generate artifacts.

> Although a mention of the general complexity of their method is shown, it would be very much helpful for the reader to display the computational time for the authors technique – along with the benchmark methods.

Time complexity is defined as <number of patches> times <time complexity of the deep learning model> and thus it will vary from model to model, computer to computer and so on. Using overlapping windows increases the number of patches by a factor of 4 and increases the complete time complexity by the same factor.

> The SSIM should be presented in detail and not just put in a reference other established accuracy metrics (such as overall accuracy, precision, recall and F1 score) be also fruitful?

Agreed. We added a new metric: the DICE coefficient, which corresponds to the F1 score.

REVIEWER #2

> Eq 6 to Eq 10 are not clear enough, especially the definition of w(i).

The text was modified to clarify and provide a better explanation of the equations.

It is necessary to add commonly used metrics for segmentation tasks, such as DICE.

Agreed, we added two more experiments (see fig. 3).

> Eq 10, should be if i <= L/2 and j <= J/2.

Corrected.

REVIEWER #3

> Was a different color look-up table used for the pyramidal window? […] Was it necessary to compensate for this different integrated area by using a different color look-up table, or alternatively, were the data normalized prior to visualizing as a 2D heatmap?

The same colour look-up tables were used in the article. The un-normalized windows were normalized.

> The description of the different approaches and different methods is a bit confusing on page 5, for the top 3 paragraphs. For example, the first paragraph refers to "The five different approaches", while the 3rd paragraph (which is the Fig. 3 caption) refers to "the adjusted SSIM for six different methods". It is a bit unclear of what an "approach" is vs. a "method" and whether there are 5 or 6 of them.

Changed all occurrences of “approach” to “method”, and 5 to 6. There are 5 methods plus 1 baseline, clarified in the revised article.

> In Fig. 3a, there appear to be 5 different window functions shown. For similar reasons, the 2nd paragraph on page 5 is also confusing, in the sentence that reads "Due to a low variance in the SSIM indices for each method but a high variance between methods..." It is unclear here as well what the method is that the sentence is referring to. Clearing up the language in these 3 paragraphs is needed. The authors might help to clarify the vertical axis of Figure 3 by explaining that this is the improvement in SSIM over baseline.

The three paragraphs were partially reformulated.

> On page 6, first paragraph of the Discussion section, the 2nd sentence contains the word "theory" twice.

Corrected.

---

## [Decision Letter · Decision Letter 1]

27 Jan 2020

PONE-D-19-29003R1

Introducing Hann windows for reducing edge-effects in patch-based image segmentation

PLOS ONE

Dear Mr. Pielawski,

Thank you for submitting your manuscript to PLOS ONE. After careful consideration, we feel that it has merit but does not fully meet PLOS ONE’s publication criteria as it currently stands. Therefore, we invite you to submit a revised version of the manuscript that addresses the points raised during the review process.

Please address the comments of reviewer 2. 

We would appreciate receiving your revised manuscript by Mar 12 2020 11:59PM. To enhance the reproducibility of your results, we recommend that if applicable you deposit your laboratory protocols in protocols.io, where a protocol can be assigned its own identifier (DOI) such that it can be cited independently in the future. For instructions see: http://journals.plos.org/plosone/s/submission-guidelines#loc-laboratory-protocols

We look forward to receiving your revised manuscript.

Kind regards,

Jie Zhang

Academic Editor

PLOS ONE

Reviewers' comments:

Reviewer's Responses to Questions

**Comments to the Author**

1. If the authors have adequately addressed your comments raised in a previous round of review and you feel that this manuscript is now acceptable for publication, you may indicate that here to bypass the “Comments to the Author” section, enter your conflict of interest statement in the “Confidential to Editor” section, and submit your "Accept" recommendation.

Reviewer #1: All comments have been addressed

Reviewer #2: All comments have been addressed

2. Is the manuscript technically sound, and do the data support the conclusions?

Reviewer #1: Yes

Reviewer #2: Yes

3. Has the statistical analysis been performed appropriately and rigorously? 

Reviewer #1: Yes

Reviewer #2: Yes

4. Have the authors made all data underlying the findings in their manuscript fully available?

Reviewer #1: Yes

Reviewer #2: Yes

5. Is the manuscript presented in an intelligible fashion and written in standard English?

Reviewer #1: Yes

Reviewer #2: Yes

6. Review Comments to the Author

Reviewer #1: (No Response)

Reviewer #2: The author showed the experiment result of Dice in Fig 4, but didn't give the overall performance comparision in terms of Dice and the discussion is not sufficient.

7. PLOS authors have the option to publish the peer review history of their article (what does this mean?). If published, this will include your full peer review and any attached files.

Reviewer #1: No

Reviewer #2: No

---

## [Author Response · Author response to Decision Letter 1]

14 Feb 2020

We thank the reviewers for their constructive comments and have revised the manuscript to address their concerns.

> The author showed the experiment result of Dice in Fig 4, but didn't give the overall performance comparison in terms of Dice

We added a new figure (fig 5), containing statistical testing of the different method with respect to the baseline and triangular/cui.

> The discussion is not sufficient.

The discussion section was modified and some content was added.

---

## [Editor Report · Decision Letter 2]

18 Feb 2020

Introducing Hann windows for reducing edge-effects in patch-based image segmentation

PONE-D-19-29003R2

Dear Dr. Pielawski,

We are pleased to inform you that your manuscript has been judged scientifically suitable for publication and will be formally accepted for publication once it complies with all outstanding technical requirements.

With kind regards,

Jie Zhang

Academic Editor

PLOS ONE

Additional Editor Comments (optional):

The authors have adequately addressed the reviewers' comments and the revised manuscript can be accepted.
---

## [Editor Report · Acceptance letter]

25 Feb 2020

PONE-D-19-29003R2 

Introducing Hann windows for reducing edge-effects in patch-based image segmentation 

Dear Dr. Pielawski:

I am pleased to inform you that your manuscript has been deemed suitable for publication in PLOS ONE. Congratulations! Your manuscript is now with our production department. 

With kind regards,

on behalf of

Dr. Jie Zhang 

Academic Editor

PLOS ONE